# Kinetics of Quality Changes in Soaking Water during the Retting Process of Pepper Berries (*Piper nigrum* L.)

**Puteri Nurain Megat Ahmad Azman** [1]**, Rosnah Shamsudin** [1,2,]*****, Hasfalina Che Man** [3] **and Mohammad Effendy Ya'acob** [1]

[1]   Department of Process and Food Engineering, Faculty of Engineering, Universiti Putra Malaysia, Serdang 43400, Malaysia; gs53584@student.upm.edu.my (P.N.M.A.A.); m_effendy@upm.edu.my (M.E.Y.)
[2]   Institute of Advanced Technology, Universiti Putra Malaysia, Serdang 43400, Malaysia
[3]   Department of Biological and Agricultural Engineering, Faculty of Engineering, Universiti Putra Malaysia, Serdang 43400, Malaysia; hasfalina@upm.edu.my
*****   Correspondence: rosnahs@upm.edu.my; Tel.: +60-03-97696366

**Abstract:** There are organic matters and bioactive compounds naturally present in pepper that may have leached out into the soaking water due to a prolonged retting process that caused changes in water quality. This study was carried out to determine the influences of different quantities of mature pepper berries and soaking time on the quality of soaking water during the retting process. The soaking test was conducted by having three tanks with different quantities of pepper berries soaked in the 18 L of water. The results show that the tank with the highest quantities of pepper berries has the highest increment of turbidity from 21.80 ± 1.90 NTU to 1103.30 ± 23.10 NTU (98%), the highest reduction in pH from 6.99 ± 0.02 to 3.67 ± 0.02 (47.50%), the highest reduction in dissolved oxygen from 5.19 ± 0.17 mg/L to 1.05 ± 0.02 mg/L (79.77%) and the highest increment of chemical oxygen demand from 21.67 ± 1.15 mg/L to 3243.33 ± 5.77 mg/L (99.33%) compared to other tanks. Furthermore, the zero, first and second-order kinetic models fitted well with the experimental data of the quality of soaking water for three conditions using the Arrhenius law approach. Thus, these findings are useful for estimating water quality during the retting process.

**Keywords:** pepper; retting; soaking water; kinetics; quality changes

## 1. Introduction

Pepper (*Piper nigrum* L.) is known as the 'King of Spices' because it is one of the most popular and oldest spices in the world. It has a sharp, pungent aroma and flavour and is also light in colour [1]. In Malaysia, 98% of total black pepper production is by the largest pepper producing state, Sarawak, while the remaining 2% is produced by other states such as Johor [2]. Pepper has various types, including green, yellow or red, black and white peppers. The green pepper berries are the immature berries, meanwhile the yellow or the red pepper berries are the fully mature berries. Common types of pepper that are very well known by people are white pepper and black pepper. According to the Malaysian Pepper Board (MPB) [3], there are few recommended local varieties of pepper such as Kuching, Semengok Emas and Semengok Aman. The Kuching pepper is the most widely grown cultivar in Sarawak and Johor when compared to other varieties. It has vigorous growth and a higher yield. Kuching is a more preferred variety for white pepper production due to the denser canopy and thinner pericarp than others.

During the production of white pepper, soaking is an essential and necessary retting process to soften the pericarp of mature pepper berries. In Sarawak, the current method used in the production of white pepper is to soak the fresh pepper berries for 12–14 days under running water, such as rivers after

harvesting and threshing [2]. However, the soaking process in Johor is performed by using stagnant water, which will be changed regularly. According to Aziz et al. [4] and Mazaheri and Mozaffari [5], there are possibilities that the organic matters and bioactive compounds that are naturally present in pepper may have leached out into the soaking water due to a prolonged retting process. At the same time, soaking water also contains inorganic matter and microorganisms [5]. Appropriate amounts of optimal-quality water sources are a requirement for economic development, ecological integrity and green environment.

Many stresses affect water quality, including anthropogenic activities (agricultural, urban and industrial activities) and increased use of water resources [6–9]. According to Safwat [10] and Chindaprasirt and Rattanasak [11], wastewater from industrial processes should be under controlled conditions, such as reducing the concentration of COD and neutral before discharged. In addition, there are the pH and chemical oxygen demand (COD) standards of discharge limit according to the Department of Environment Malaysia (DOE), as shown in Table 1.

**Table 1.** Standards of discharge limit for pH and chemical oxygen demand (COD).

| Parameter | Unit | DOE Standard |
|---|---|---|
| pH | - | 6–9 |
| Chemical oxygen demand (COD) | mg/L | 120 |

The determination of water qualities such as turbidity, pH, dissolved oxygen (DO) and chemical oxygen demand (COD) are essential for the further process.

The kinetic orders, such as the zero, first and second order of reactions, are used to determine the prediction of changes. The rate equation for zero, first and second order show the effect mathematically. Therefore, orders of reaction are a part of the rate equation.

To the best of the authors' knowledge, there is a lack of findings in relation to the properties of soaking water during the retting process of pepper berries. Therefore, this study aimed to (1) determine the influences of different quantities of mature pepper berries and soaking time on the quality of soaking water during the retting process and (2) develop kinetic models that explain the changes in the quality of soaking water. Thus, this study is important as it provides useful information to determine the quality of water, which can be applied in future.

## 2. Materials and Methods

### 2.1. Sample Preparations

The samples of the fresh mature pepper berries (Kuching variety) were selected and obtained from a farm in Johor, Malaysia. A locally made thresher was used to thresh the mature pepper berries and also to remove the leaves and the spikes. After that, the mature pepper berries were transported to the laboratory under an ambient temperature. Next, they were sorted by selecting only the reddish-orange or the light yellowish berries for white pepper processing. The sorted mature pepper berries were stored in the chiller (TD-1600, PROTECH, Malaysia) overnight with a temperature of 3–5 °C.

### 2.2. Soaking Test

There were three different tanks (A, B and C) and six samples would be used for the soaking test. Each sample was weighed as 1 kg. Then, each jute gunny bag was filled with 1 kg of pepper berries and its dimensions were 28 cm (length), 23 cm (width) and 2.5 cm (height). For this soaking test, the 1 kg gunny (jute) bag filled with pepper berries and tap water were placed into different tanks. In tank A, 1 kg of pepper berries in a jute gunny bag was used. Meanwhile, 2 kg of pepper berries (2 jute gunny bag) were used for tank B and 3 kg of pepper berries (3 jute gunny bag) were used for tank C. However, these three tanks used the same amount of water, which was 18 L. Figure 1A–C show the dimensions of the tank, gunny bag and height of water.

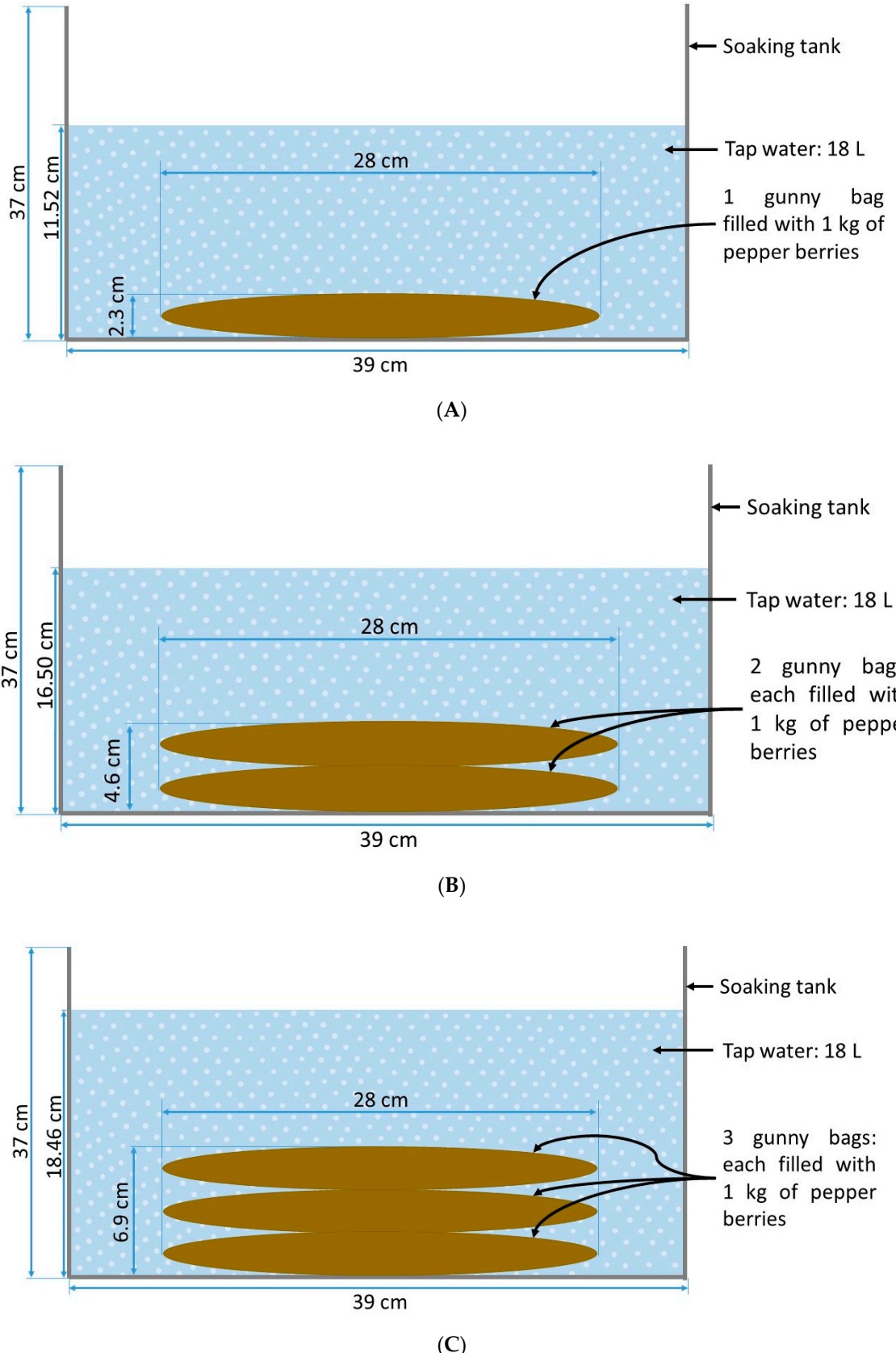

**Figure 1.** Drawing of tanks with different quantities of pepper berries soaked in water: (**A**) a tank of 1 gunny bag; (**B**) a tank of 2 gunny bags; (**C**) a tank of 3 gunny bags. Each tank was filled with 1kg of pepper berries, soaked in 18 L water.

The proportion of pepper berries to tap water during the pepper berries soaking test was 1:6 [12]. The tank used during soaking was transparent polypropylene tank with a length of 39 cm, a width of 25 cm and a depth of 37 cm. The soaking test in three tanks was monitored for 7 days without changing the water and 100 mL of soaking water was sampled every day to determine the properties of soaking water during soaking. The sample of soaking water for the day 0 shall be taken as soon as all the pepper berries in the prescribed conditions have been soaked.

## 2.3. Analysis of Soaking Water Quality Parameters

### 2.3.1. Determination of Turbidity

About 10 mL of the water sample was used to determine water clarity. It is the expression of the light amount dispersed by any substance in the water when light passes through the water sample. The parameter of turbidity was determined using a turbidity meter (2100 Q, HACH, USA).

### 2.3.2. Determination of pH

pH is a scale to determine the acidity, the neutral or the alkalinity of a water sample. A pH meter (Spear pH Tester, China) was used to record the pH of soaking water at room temperature.

### 2.3.3. Determination of Dissolved Oxygen (DO)

DO is expressed as the oxygen level that dissolves in the water. DO concentration was measured by using an instrument of YSI Professional Plus Multimeter (Xylem, USA).

### 2.3.4. Determination of Chemical Oxygen Demand (COD)

COD is expressed as a measurement of the amount of pollutant in the water which cannot be oxidized biologically. Using a spectrophotometer (DR/4000U, HACH, USA), the COD value of soaking water was measured.

## 2.4. Statistical Analysis and Kinetic Models

A Tukey's test was performed for all data as a function of time using the Minitab Statistic 16 Edition to differentiate and to determine the significance between the mean values. Each analysis was done by having triplicate data, and the mean and the standard deviation were stated. The confidence limits were considered as 95% ($p < 0.001$). Using the Pearson correlation coefficient ($p < 0.05$), the correlations among each measurement such as turbidity, pH, DO and COD were evaluated. Statistical analyses were performed using Minitab Statistic 16 Edition. The kinetic of the quality changes observed for soaking water during white pepper production was described by zero, first and second-order kinetic models. The most accurate kinetic model-fitting analysis was evaluated based on the determination coefficient ($R^2$). The order of the reaction was conducted at various parameter of water quality depending on the soaking time. The zero, first and second-order kinetic models were expressed as Equations (1)–(3).

$$Zero\ order = C = -kt + C_0 \tag{1}$$

$$First\ order = \ln C = -kt + \ln C_0 \tag{2}$$

$$Second\ order = \frac{1}{C} = kt + \frac{1}{C_0} \tag{3}$$

where $C$ = the measured value for each parameter of water quality; $C_0$ = the initial value of the measured parameter of water quality; $k$ = the rate constant; $t$ = the soaking time [13].

## 3. Results and Discussions

Properties of soaking water quality is one of the most important factors that affect the quality of white pepper. Tables 2 and 3 show the properties of soaking water quality consisting of turbidity, pH, DO and COD during white pepper production in different tanks (tanks A, B and C).

**Table 2.** Mean values for turbidity and pH of soaking water during 7 days of soaking test.

| Day | Properties of Soaking Water | | | | | |
| --- | --- | --- | --- | --- | --- | --- |
| | Turbidity (NTU) | | | pH | | |
| | A | B | C | A | B | C |
| 0 | 18.67 ± 4.23 [g] | 20.10 ± 4.20 [h] | 21.80 ± 1.90 [h] | 7.14 ± 0.04 [a] | 7.07 ± 0.04 [a] | 6.99 ± 0.02 [a] |
| 1 | 47.70 ± 3.27 [f] | 55.30 ± 0.70 [g] | 82.20 ± 1.20 [g] | 6.18 ± 0.02 [b] | 5.46 ± 0.02 [b] | 5.44 ± 0.01 [b] |
| 2 | 73.43 ± 9.72 [f] | 85.80 ± 3.10 [f] | 178.70 ± 29.20 [f] | 5.98 ± 0.02 [c] | 5.26 ± 0.02 [c] | 5.26 ± 0.03 [c] |
| 3 | 252.33 ± 7.23 [e] | 424.70 ± 7.60 [e] | 499.00 ± 7.50 [e] | 5.09 ± 0.03 [d] | 5.14 ± 0.07 [d] | 4.83 ± 0.04 [d] |
| 4 | 450.67 ± 7.57 [d] | 605.70 ± 7.10 [d] | 667.70 ± 2.10 [d] | 4.46 ± 0.04 [e] | 4.53 ± 0.02 [e] | 4.43 ± 0.02 [e] |
| 5 | 508.00 ± 5.29 [c] | 752.70 ± 11.40 [c] | 866.70 ± 5.80 [c] | 4.28 ± 0.03 [f] | 4.31 ± 0.01 [f] | 4.22 ± 0.01 [f] |
| 6 | 642.67 ± 11.24 [b] | 930.00 ± 20.00 [b] | 1040.00 ± 10.00 [b] | 4.08 ± 0.03 [g] | 4.05 ± 0.01 [g] | 3.91 ± 0.01 [g] |
| 7 | 870.00 ± 20.00 [a] | 1023.30 ± 15.30 [a] | 1103.30 ± 23.10 [a] | 3.96 ± 0.01 [h] | 3.84 ± 0.02 [h] | 3.67 ± 0.02 [h] |

Data are expressed mean ± SD; A, tank of 1 gunny bag; B, tank of 2 gunny bags; C, tank of 3 gunny bags; each filled with 1 kg of pepper berries, soaked in 18 L water. Different letters indicate statistically significant differences exist $p < 0.001$ for each column. Means do not share a letter are significantly different. Tukey's test was applied with 95% simultaneous confidence intervals.

**Table 3.** Mean values for dissolved oxygen (DO) and COD of soaking water during 7 days of soaking test.

| Day | Properties of Soaking Water | | | | | |
| --- | --- | --- | --- | --- | --- | --- |
| | DO (mg/L) | | | COD (mg/L) | | |
| | A | B | C | A | B | C |
| 0 | 5.23 ± 0.21 [a] | 5.20 ± 0.19 [a] | 5.19 ± 0.17 [a] | 8.67 ± 1.15 [h] | 18.33 ± 0.58 [h] | 21.67 ± 1.15 [h] |
| 1 | 1.96 ± 0.14 [b] | 1.91 ± 0.05 [b] | 1.89 ± 0.14 [b] | 217.33 ± 1.53 [g] | 313.33 ± 5.69 [g] | 420.67 ± 2.08 [g] |
| 2 | 1.85 ± 0.04 [bc] | 1.78 ± 0.05 [bc] | 1.63 ± 0.02 [c] | 510.00 ± 2.65 [f] | 720.67 ± 0.58 [f] | 854.00 ± 2.65 [f] |
| 3 | 1.67 ± 0.04 [cd] | 1.58 ± 0.04 [cd] | 1.43 ± 0.03 [cd] | 966.67 ± 5.77 [e] | 1205.67 ± 1.15 [e] | 1384.33 ± 5.13 [e] |
| 4 | 1.48 ± 0.03 [de] | 1.45 ± 0.02 [de] | 1.34 ± 0.01 [de] | 1004.00 ± 4.00 [d] | 1265.67 ± 6.03 [d] | 1796.67 ± 5.77 [d] |
| 5 | 1.35 ± 0.04 [ef] | 1.30 ± 0.03 [ef] | 1.26 ± 0.05 [def] | 1190.33 ± 4.16 [c] | 1310.00 ± 2.65 [c] | 2186.67 ± 5.77 [c] |
| 6 | 1.27 ± 0.02 [ef] | 1.25 ± 0.02 [ef] | 1.15 ± 0.04 [ef] | 1686.67 ± 5.77 [b] | 1706.67 ± 5.77 [d] | 2576.67 ± 2.89 [b] |
| 7 | 1.19 ± 0.02 [f] | 1.11 ± 0.02 [f] | 1.05 ± 0.02 [f] | 2133.33 ± 5.77 [a] | 2616.67 ± 5.77 [a] | 3243.33 ± 5.77 [a] |

Data are expressed mean ± SD; DO, dissolved oxygen (mg/L); COD, chemical oxygen demand (mg/L); A, tank of 1 gunny bag; B, tank of 2 gunny bags; C, tank of 3 gunny bags; each filled with 1kg of pepper berries, soaked in 18 L water. Different letters indicate statistically significant differences exist $p < 0.001$ for each column. Means do not share a letter are significantly different. Tukey's test was applied with 95% simultaneous confidence intervals.

### 3.1. Determination of Turbidity

The initial turbidity of water in tank A was 18.67 ± 4.23 NTU, as shown in Table 1. The turbidity of water in tank A has increased sharply with the value of 47.70 ± 3.27 NTU on the first day, the increment of which was 60.86% from the initial. Then, it has increased gradually until it reached 870 ± 20 NTU at day 7 with an increment of 97.85%. Next, tank B has the initial value of turbidity which was 20.13 ± 4.20 NTU. The turbidity of water attained on the first day in tank B was 55.33 ± 0.70 NTU with the increment by 63.62%. On the 7th day of soaking, it increased gradually by 98% with the value of 1023.33 ± 15.20 NTU. Furthermore, the initial value for the turbidity of water in tank C

was 21.80 ± 1.90 NTU. It increased sharply after a day of soaking with the value of 82.20 ± 1.20 NTU (73.48%). After that, it reached 1103.30 ± 23.10 NTU on day 7 of soaking with an increment of 98%. Therefore, the longer of soaking time, the higher the turbidity of soaking water. These increments in turbidity during soaking were similar to previous work on soaking of soybean [12].

As can be seen, the turbidity of soaking water in all three tanks of A, B and C increased gradually from the first day until it reached the highest value on the 7th day of soaking. These increments in the turbidity of soaking water were affected by quantity of pepper berries. The increment in turbidity of soaking water on the 7th day between tanks A and B was 14.98%; meanwhile, the increment between tanks B and C was 7.25%. Therefore, the quantity of pepper berries also influenced the turbidity of soaking water, as shown in Table 2. From the observation during the soaking test, the colour of soaking water in all three tanks had changed to a dark colour.

Overall, the turbidity of soaking water was affected significantly by the time ($p < 0.001$) in tanks A, B and C. The initial turbidity values were found to be 18.67 NTU, 20.13 NTU and 21.80 NTU in tanks A, B and C, respectively. On the first day of soaking, the turbidity value had changed rapidly (47.70 NTU, 55.33 NTU and 82.20 NTU for tanks A, B and C) and then increased steadily at day 2 until day 7. The turbidity of soaking water at day 2 reached 73.43 NTU, 85.80 NTU, and 178.70 NTU in tanks A, B and C, and increased until 870 NTU, 1023.33 NTU, 1103.30 NTU in tanks A, B and C, respectively, on day 7. The increment of turbidity values during soaking for tank A was 97.85%; this was 98% in both tanks B and C. The soaking water in tank C showed the highest turbidity. This highlights the increment of turbidity in these tanks due to the leaching of hydrolyzable tannin in the soaking water and produced undesirable dark brown colour [14,15]. It means that the water discharged from soaking was rich in organic matter, including hydrolyzable tannin (gallotannin or common tannic acid) yielding a dark tan colour. Therefore, the turbidity of soaking water was higher when a large quantity of pepper berries soaked in water due to more leaching of hydrolyzable tannin and a dark colour being produced. The turbidity of the soaking water is a vital parameter that might be used to reach a decision for the next soaking operation by reusing the water and also to prevent environmental problems caused by discharging the water [12]. Thus, the changes in turbidity in tank C, which had the largest quantity of pepper berries soaked in water during 7 days of soaking was the highest turbidity.

### 3.2. Determination of pH

Based on Table 2, the time affected the pH of soaking water significantly at $p < 0.001$. In tank A, the initial value of pH was 7.14 ± 0.04. The pH of water attained on the first day in tank A was 6.18 ± 0.02 with the reduction by 13.45%. On the 7th day of soaking, it decreased gradually by 44.54% with the value of 3.96 ± 0.01. Next, the initial pH of the water for tank B was 7.07 ± 0.04. Table 2 shows the pH of water in tank B had decreased sharply with the value of 5.46 ± 0.02 on the first day, which its reduction was 22.77% from the initial. Then, it decreased gradually until it reached to 3.84 ± 0.02 on day 7 with the reduction by 45.69%. Besides, the initial value for pH of water in tank C was determined as 6.99 ± 0.02. It decreased sharply after a day of soaking with a value of 5.44 ± 0.01 (22.17%). After that, it gradually decreased until it reached 3.67 ± 0.02 on day 7 of soaking with the reduction by 47.50%. Therefore, the longer the soaking time, the lower the pH of soaking water. These reductions in pH during soaking was similar to previous work on the soaking of soybeans [12].

As indicated, the pH of soaking water in all three tanks of A, B and C had decreased gradually from the first day until they reached the lowest value on the 7th day of soaking. These reductions in the pH of soaking water were affected by the quantity of pepper berries. The reduction in the pH of soaking water on the 7th day between tanks A and B was 3.03%; meanwhile, tanks B and C had a reduction of 4.43%. Therefore, the quantity of pepper berries also influenced the pH of soaking water, as shown in Table 2, even though the reduction between these tanks was not high.

Overall, the pH of the soaking water had decreased gradually concerning time. The change in pH values leading to acidity can be observed and used to predict the rate of leaching acidic compounds from pepper berries into the water. Consequently, it helps to understand the physical

and the chemical changes in the water. The initial pH of the soaking water was 7.14, 7.07 and 6.99 in tanks A, B and C, which were close to neutral. However, the pH value had changed rapidly during the first day (6.18, 5.46 and 5.44 for tanks A, B and C, respectively) and then decreased steadily on day 2 until day 7. Table 2 shows that the pH of soaking water on day 2 gradually reached to 5.98, 5.26 and 5.26 in tanks A, B and C, respectively. The pH values continued to decrease until 3.96, 3.84 and 3.67 in tanks A, B and C, respectively, on day 7. Therefore, the reduction in pH values during soaking for tank A was by 44.54%, 45.69% in tank B and 47.50% in tank C. Among these 3 tanks, the pH value in tank C was the lowest indicating the highest acidity due to the increasing acidic compounds leached from pepper berries in water during 7 days of soaking. Moreover, the large quantity of pepper berries soaked in water affected the acidity value in tank C. Therefore, the soaking water should be treated according the proper standard to prevent the pollution.

### 3.3. Determination of Dissolved Oxygen (DO)

Change in DO during soaking in different conditions is shown in Table 3. In tank A, the initial DO of water was $5.23 \pm 0.21$ mg/L. Table 3 shows that the DO of water in tank A had decreased sharply with the value of $1.96 \pm 0.14$ mg/L on the first day, the reduction in which was 62.52% from the initial. Then, it decreased gradually until it reached $1.19 \pm 0.02$ mg/L at day 7 with the reduction of 77.25%. Next, tank B had the initial value of DO which was $5.20 \pm 0.19$ mg/L. The DO of water attained on the first day in tank B was $1.91 \pm 0.05$ mg/L with a sharp reduction of 63.27%. On the 7th day of soaking, it had decreased gradually by 78.65% with the value of $1.11 \pm 0.02$ mg/L. Additionally, the initial value for DO of water in tank C was measured as $5.19 \pm 0.17$ mg/L. It had decreased sharply after a day of soaking with a value of $1.89 \pm 0.14$ mg/L (63.58%). After that, it reached $1.05 \pm 0.02$ mg/L on day 7 of soaking with the reduction of 79.77%. Therefore, the longer the soaking time, the higher the DO of soaking water.

As indicated, the DO of soaking water in all three tanks (A, B and C) had decreased gradually from the first day until it reached the lowest value on the 7th day of soaking. These reductions in the DO of soaking water were affected by pepper berries. The reduction in the DO of soaking water on the 7th day between tanks A and B was 6.72%; meanwhile, tanks B and C had a reduction of 5.41%. Therefore, the quantity of pepper berries also influenced the DO of soaking water, as shown in Table 3, even though the reduction between these tanks were not high.

Overall, the DO of water on the first day in all tanks had decreased sharply (1.96, 1.91 and 1.89 mg/L in tanks A, B and C) compared to the initial value of DO (5.23, 5.20 and 5.19 mg/L in tanks A, B and C). Due to the prolonged soaking of pepper berries in water, bacteria break down organic matter naturally, which consumes some oxygen in the process [4]. The oxygen can be reduced to the degree that it can absorb the water organism when the decomposition of organic matter is high. However, DO values in tanks A, B and C changed slightly on day 2 (1.85, 1.78 and 1.63 mg/L) and decreased steadily until day 7 (1.05 mg/L for tank A, 1.19 mg/L for tank B and 1.11 mg/L for tank C). The values of DO on day 7 for the three tanks were lower, which had caused anaerobic fermentation and consequently the produced biogas. According to Suthersan and McDonough [16], the DO value in the anaerobic condition is often less than 0.5 due to a lack of free oxygen, which was induced by natural contamination. Thus, the reduction in DO values during soaking for tank A by 77.25%, 78.65% in tank B and 79.77% in tank C were affected significantly by time ($p < 0.001$).

### 3.4. Determination of Chemical Oxygen Demand (COD)

The amount of organic and inorganic oxydizable compounds in water can be determined by measuring the COD value [17]. In tank A, the initial value of COD was $8.67 \pm 1.15$ mg/L. The COD of water attained on the first day in tank A was $217.33 \pm 1.53$ mg/L with the sharp increment of 96.01%. On the 7th day of soaking, it had increased gradually to 99.59% with the value of $2133.33 \pm 5.77$ mg/L. Next, the initial COD of water in tank B was $18.33 \pm 0.58$ mg/L. Table 2 shows that the COD of water in tank B had increased sharply with the value of $313.33 \pm 5.69$ mg/L on the first day, the increment of

which was 94.15% from the initial. Then, it increased gradually until it reached 2616.67 ± 5.77 mg/L on day 7 with the increment by 99.30%. Furthermore, the COD of water in tank C had the initial value of 21.67 ± 1.15 mg/L. It increased sharply after a day of soaking with a value of 420.67 ± 2.08 mg/L (94.85%). After that, it reached 3243.33 ± 5.77 mg/L on day 7 of soaking with the increment by 99.33%. Therefore, the longer the soaking time, the higher the COD of soaking water.

As can be seen, the COD of soaking water in all three tanks of A, B and C had increased gradually from the first day until they reached the highest value on the 7th day of soaking. These increments in the COD of soaking water were affected by pepper berries. The increment in the COD of soaking water on the 7th day between tanks A and B was 18.47%; meanwhile, tanks B and C had an increment of 19.32%. Therefore, the quantity of pepper berries has also affected the COD of soaking water, as shown in Table 3.

Overall, the COD of soaking water showed a significant increment ($p < 0.001$) that was affected by time. The initial value of COD in tanks A, B and C were 8.67, 18.33 and 21.67 mg/L and tank C was the highest when compared to other tanks. COD of soaking water on the first day had sharply increased in tanks A, B and C with the values of 217.33, 313.33 and 420.67 mg/L, respectively. Then, COD gradually increased from day 2 (510, 720.67 and 854 mg/L for tanks A, B and C) until day 7 (2133.33 mg/L, 2616.67 and 3243.33 mg/L for tanks A, B and C). Based on the previous work of Tharmalingam et al. [14], the value of COD obtained was 5757 mg/L, which was higher than the COD value measured on day 7 in this study. This difference may be caused by the preparation of materials and the soaking system used. The increment of COD during soaking for tank A was 99.59%, 99.30% in tank B and 99.33% in tank C. Thus, tank C has the highest value of COD when compared to other tanks, even though the increment in COD of soaking water in tank A is the highest.

### 3.5. Correlation of Water Quality

The correlation analysis revealed a significant ($p < 0.05$) water quality attributes in soaking water, as shown in Table 4 for 3 conditions (A, B and C). For condition A, the correlation coefficient between turbidity and chemical oxygen demand (COD), and pH and DO were highly positive, with values of 0.97 and 0.82, respectively. This study indicates that turbidity increased proportionally with the increase in COD, and the same applies to the decrease in pH proportionally to the decrease in dissolved oxygen (DO). Meanwhile, the correlation coefficients between pH, turbidity and COD were highly negative with values of −0.91 and −0.92, respectively. The result reveals that the pH decreased proportionally with the increase in turbidity and COD. However, the lowest coefficient of correlation is related to the relationships between DO, turbidity and COD which were −0.61 and −0.68.

**Table 4.** Correlation on water quality attributes (turbidity, pH, DO and COD).

| Condition | A | | | | B | | | | C | | | |
|---|---|---|---|---|---|---|---|---|---|---|---|---|
| Parameter | Turbidity (NTU) | pH | DO (mg/L) | COD (mg/L) | Turbidity (NTU) | pH | DO (mg/L) | COD (mg/L) | Turbidity (NTU) | pH | DO (mg/L) | COD (mg/L) |
| Turbidity (NTU) | 1 | | | | 1 | | | | 1 | | | |
| pH | −0.91 ** | 1 | | | −0.87 ** | 1 | | | −0.90 ** | 1 | | |
| DO (mg/L) | −0.61 ** | 0.82 ** | 1 | | −0.63 ** | 0.91 ** | 1 | | −0.65 ** | 0.90 ** | 1 | |
| COD (mg/L) | 0.97 ** | −0.92 ** | −0.68 ** | 1 | 0.93 ** | −0.88 ** | −0.70 ** | 1 | 0.98 ** | −0.93 ** | −0.71 ** | 1 |

Data are expressed DO, dissolved oxygen (mg/L); COD, chemical oxygen demand (mg/L); A, tank of 1 gunny bag; B, tank of 2 gunny bags; C, tank of 3 gunny bags; each filled with 1kg of pepper berries, soaked in 18 L water. ** Correlations are significant at $p < 0.05$.

For condition B, the correlation coefficient between turbidity and chemical oxygen demand (COD), and pH and DO were highly positive with the value of 0.93 and 0.91, respectively. This study indicates that turbidity increased proportionally with the increase in COD, and the same applies to the decrease in pH proportionally to the decrease in dissolved oxygen (DO). Meanwhile, the correlation coefficient

between pH, turbidity and COD were highly negative with values of −0.87 and −0.88, respectively. The result reveals that the pH decreased proportionally with the increase in turbidity and COD. However, the lowest coefficient of correlation is related to the relationships between DO, turbidity and COD, which were −0.63 and −0.70.

For condition C, the correlation coefficient between turbidity and chemical oxygen demand (COD), and pH and DO were highly positive with the value of 0.98 and 0.90, respectively. This study indicates that turbidity increased proportionally with the increase in COD, and the same applies to the decrease in pH proportionally to the decrease in dissolved oxygen (DO). Meanwhile, the correlation coefficient between pH and turbidity, COD were highly negative with the value of −0.90 and −0.93, respectively. The result reveals that the pH decreased proportionally with the increase in turbidity and COD. However, the lowest coefficient of correlation is related to the relationships between DO, turbidity and COD, which were −0.65 and −0.71.

In all cases, these results reveal that the retting processes for condition A, B and C have the same trend of correlation on water quality attributes. Various correlations between the output parameters and their coefficients can be explained with respect to the source material [18,19].

### 3.6. Kinetics of Quality Changes in Soaking Water

The kinetic parameters for the quality changes in soaking water for condition A, B and C are shown in Table 5. For turbidity, the $R^2$ values of condition A were 0.956, 0.925 and 0.650 for zero, first and second-order kinetic models, respectively. In contrast, the $R^2$ values for condition B were 0.965, 0.888 and 0.623, while condition C showed the $R^2$ values of 0.976, 0.865 and 0.525 for zero, first and second-order kinetic models, respectively. Additionally, the rate constant for turbidity in the zero, first and second-order kinetic models increased from −124.240, −161.64 and −173.72 (day$^{-1}$) to −0.551, −0.577 and −0.538 (day$^{-1}$) and increased continuously until 0.006, 0.006 and 0.005 for condition A, B and C, thereby indicating the high influence in turbidity of the soaking water which decreased in time during retting process.

The pH based on the kinetic parameters was in agreement with the second-order kinetic model based on the higher $R^2$ values obtained compared to the values for the zero and first-order kinetic model (Table 5) for condition A, B and C. The pH for condition A was indicated by $R^2$ values of 0.928, 0.952 and 0.971 for zero, first and second-order kinetic models, respectively. On the other hand, condition B obtained $R^2$ values of 0.868, 0.923 and 0.959, while condition C achieved the $R^2$ values of 0.887, 0.942 and 0.975 for the zero, first and second-order kinetic models, respectively. Additionally, the rate constant of pH in the zero, first and second-order kinetic models decreased from 0.458, 0.394 and 0.410 (day$^{-1}$) to 0.087, 0.077 and 0.082 (day$^{-1}$) and decreased continuously until −0.033, −0.016 and −0.017 for condition A, B and C.

In contrast, the $R^2$ values obtained for DO were 0.534, 0.530 and 0.521 (zero-order model), 0.704, 0.705 and 0.696 (first-order model) and 0.877, 0.882 and 0.878 (second-order model) for condition A, B and C, respectively. Additionally, the rate constant for DO in the zero, first and second-order kinetic models decreased from 0.398, 0.399 and 0.403 (day$^{-1}$) to 0.162, 0.166 and 0.173 (day$^{-1}$) and decreased continuously until −0.079, −0.084 and −0.091 for condition A, B and C.

COD also indicated that the $R^2$ values of the zero, first and second-order kinetic models for condition A were 0.971, 0.701 and 0.359 (Table 5). For COD in condition B, the values of $R^2$ were 0.928, 0.692 and 0.370 for zero, first and second-order kinetic models, respectively. On the other hand, condition C obtained $R^2$ values of 0.996, 0.711 and 0.367 for the zero, first and second-order kinetic models, respectively. Additionally, the rate constant of COD in the zero, first and second-order kinetic models increased from −289.26, −321.23 and −449.31 (day$^{-1}$) to −0.612, −0.536 and −0.562 (day$^{-1}$) and increased continuously until 0.010, 0.005 and 0.004 for condition A, B and C.

**Table 5.** Kinetics on quality changes of soaking water in condition A, B and C.

| Parameter | Condition | Zero-Order Model | | First-Order Model | | Second-Order Model | |
|---|---|---|---|---|---|---|---|
| | | $k$ (NTUday$^{-1}$) | $R^2$ | $k$ (day$^{-1}$) | $R^2$ | $k$ (NTU$^{-1}$day$^{-1}$) | $R^2$ |
| Turbidity | A | −124.24 | 0.956 | −0.551 | 0.925 | 0.006 | 0.650 |
| | B | −161.64 | 0.965 | −0.577 | 0.888 | 0.006 | 0.623 |
| | C | −173.72 | 0.976 | −0.538 | 0.865 | 0.005 | 0.525 |
| Parameter | Condition | Zero-Order Model | | First-Order Model | | Second-Order Model | |
| | | $k$ (day$^{-1}$) | $R^2$ | $k$ (day$^{-1}$) | $R^2$ | $k$ (day$^{-1}$) | $R^2$ |
| pH | A | 0.458 | 0.928 | 0.087 | 0.952 | −0.033 | 0.971 |
| | B | 0.394 | 0.868 | 0.077 | 0.923 | −0.016 | 0.959 |
| | C | 0.410 | 0.887 | 0.082 | 0.942 | −0.017 | 0.975 |
| Parameter | Condition | Zero-Order Model | | First-Order Model | | Second-Order Model | |
| | | $k$ (mgL$^{-1}$day$^{-1}$) | $R^2$ | $k$ (day$^{-1}$) | $R^2$ | $k$ (day$^{-1}$) | $R^2$ |
| DO | A | 0.398 | 0.534 | 0.162 | 0.704 | −0.079 | 0.877 |
| | B | 0.399 | 0.530 | 0.166 | 0.705 | −0.084 | 0.882 |
| | C | 0.403 | 0.521 | 0.173 | 0.696 | −0.091 | 0.878 |
| Parameter | Condition | Zero-Order Model | | First-Order Model | | Second-Order Model | |
| | | $k$ (mgL$^{-1}$day$^{-1}$) | $R^2$ | $k$ (day$^{-1}$) | $R^2$ | $k$ (mg$^{-1}$Lday$^{-1}$) | $R^2$ |
| COD | A | −289.26 | 0.971 | −0.612 | 0.701 | 0.010 | 0.359 |
| | B | −321.23 | 0.928 | −0.536 | 0.692 | 0.005 | 0.370 |
| | C | −449.31 | 0.996 | −0.562 | 0.711 | 0.004 | 0.367 |

Data are expressed DO, dissolved oxygen (mg/L); COD, chemical oxygen demand (mg/L); A, tank of 1 gunny bag; B, tank of 2 gunny bags; C, tank of 3 gunny bags; each filled with 1 kg of pepper berries, soaked in 18 L water; $k$, rate constant (day$^{-1}$); $R^2$, coefficient.

## 4. Conclusions

According to the results, a soaking test for 7 days increased the turbidity in tank A from 18.67 to 870 NTU (97.85%), tank B from 20.13 to 1023.33 NTU (98%) and tank C from 21.8 to 1103.30 NTU (98%). The pH values of water reduced during soaking by 44.54% in tank A (from 7.14 to 3.96), 45.69% in tank B (from 7.07 to 3.84) and 47.50% in tank C (from 6.99 to 3.67). Due to the anaerobic fermentation that occurred during soaking, the reductions in DO values are shown in the results, which are 77.25% (from 5.23 to 1.19 mg/L) for tank A, 78.65% (from 5.20 to 1.11 mg/L) for tank B and 79.77% (from 5.19 to 1.05 mg/L) for tank C. In determination of COD, the value of soaking water increased sharply in tank A from 8.67 to 2133.33 mg/L (99.59%), tank B from 18.33 to 2616.67 mg/L (99.30%) and tank C from 21.67 to 3243.33 mg/L (99.33%). Therefore, tank C has the highest value of COD when compared to other tanks, even though the increment in COD of soaking water in tank A was the highest. Overall, tank C (3 kg of pepper berries) was the most affected by the soaking time and the quantity of pepper berries soaked in water when compared to other tanks. Thus, prolonged soaking time with A larger quantity of pepper berries significantly influenced the soaking water properties (turbidity, pH, DO and COD).

In all cases for condition A, B and C, the correlation coefficient obtained indicated that the turbidity increased proportionally with the increase in COD, and the same applies to the decrease in pH proportionally to the decrease in dissolved oxygen (DO). Meanwhile, the pH decreased proportionally with the increase in turbidity and COD.

In summary, a kinetic model that expressed the quality changes in soaking water is presented in this study. The statistical analysis of the changes in water quality showed significant differences in the analyzed parameters ($p < 0.05$). Definitely, the water quality changes in turbidity, pH, DO and COD during retting process were adequately expressed by zero, first and second-order kinetic models. The experimental data from this study is predicted to be useful for estimating water quality during retting process as well as for white pepper production sectors.

**Author Contributions:** P.N.M.A.A. conducted the experiments, collected and analyzed the data of results, and wrote the manuscript. R.S. supervised the research and revised the manuscript. H.C.M. and M.E.Y. supervised the experiments. All authors have read and agreed to the published version of the manuscript.

**Funding:** This research received no external funding.

**Acknowledgments:** The authors would like to express their appreciation to Universiti Putra Malaysia for the financial and technical support given during conducting this research work.

**Conflicts of Interest:** The authors declare no conflict of interest.

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
