# Peer review of "Kinetics of Quality Changes in Soaking Water during the Retting Process of Pepper Berries (Piper nigrum L.)"

_processes, doi:10.3390/pr8101255_

Round 1
Reviewer 1 Report
This manuscript presents potentially interesting results for readers of the journal. The experimental work seems carefully done and well destroyed in the body of the manuscript. Likewise, the results obtained satisfactorily support the conclusions. In my opinion, this manuscript can be published after minor revisions.
The minor issues I wish to point out are:
1) in section 2.4, on lines 130-131, the integrated equations of order zero and order one are presented for the kinetic processes analyzed. However, when advancing in the manuscript, adjustments of the experimental results to the integrated equation of order two are also observed. In my opinion it should also be included in section 2.4 along with those already included of order zero and order one.
2) On page 13, table 4 shows the kinetic parameters corresponding to the variation of each one of the analyzed parameters. In this case, the adjustments to the integrated equations of order zero, order one and order two are presented. It does not make sense to present the three adjustments and it is understood that the process will follow a kinetic behavior corresponding to the order that best fits. For example, for Turbidity, we obtain an R2 of 0.956 for a zero order process, 0.925 for a one order process and 0.650 for a two order process. It is evident that the variation in turbidity with time follows a process of zero order, which is the one that should be included in the table, eliminating the results of order one and order two. The same would happen for the rest of the parameters (ie PH it is evident that its variation must correspond to a process of order two)
3) In the same table 13, the units of the kinetic constants are incorrect. Only the constant of order one would have units of 1 / time, the units of the constants of order zero and order two would be different and must be corrected, in the case of order zero it would be Parameter / time and those of order two would be 1 / Parameter. Weather.
4) It would be interesting to include not only the rate constants but also the corresponding half-lives, which can be much more illustrative. Likewise, it would be interesting to include the equations of the half-lives in section 2.4 together with the integrated kinetic equations.
5) It would be convenient to include in the body of the manuscript the implications that the different kinetic orders would have for each of the parameters
Reviewer 2 Report
In the paper presented by Azman et al. the authors tried to determine the influences of different quantities of mature pepper berries and soaking time on the quality of soaking water during the retting process. They compared three different quantities and described the obtained data using the zero and first-order kinetic models. Although there are some interesting results and the area of research is interesting, some additional things need to be addressed. To be more specific:
- Abstract, rows 19-20. I suggest not talking about tank A, B and C in the abstract but just say three tanks and additionally when talking about results it would be better to say “The results showed that thank with the highest quantities of pepper barriers has the etc..”
- Abstract, row 26. Last sentence in the abstract. That goes without saying so that sentence is that sentence is redundant.
- Additional information’s should be mentioned in the introduction. First, amount of water that is usually used for retting process if there is such information. Second, authors mention water quality and water control mostly referring to COD and pH. It would be good to mention maximal concentrations that are allowed before industrial water discharge in the environment. Also, authors mention that farmers present the problem because of discharging waste water without treatment in the environment. Some assumptions about amounts would be good because usually those numbers are irrelevant in comparison to the industry.
- Materials and methods.. rows 67-73 …. text should be removed from the manuscript since it is repeating in the section sample preparation
- row 89. Authors state that the amount of water was the same in all experiments. That is not true. According to Figure 1 height of water was the same in all reactors and if gunny bags would be removed from the tanks, thank 3 would have the smallest amount of water. It’s a buoyancy law. Please recheck and add correct values of the volume in the manuscript. Additionally, this will also affect the results since some of them are expressed per litter. If it is true, all of them should be recalculated so the results can be compared.
- Figure 1. Figures should be smaller and in Figure B) and C) when marking the bags add each (2/3 gunny bags each filled with 1 kg of pepper)
- All the experiments were performed in batch experiments but the authors say that usual method during soaking process is continuous washing. Since by applying continuous soaking (i.e. river) water enters the bags and continuously rinses the berries, what is the difference in the rinsing / soaking speed and the final product in this two processes? Can the results obtained in batch experiment be copied to continuous systems?
- Table 1. When the experiments started, after what time first samples were taken? It just say 0. That should be mentioned in the method section.
- Table 3. I would suggest marking best model with bold or some other mark to make it clear which one is the best.
- Equation for second order kinetic is missing (mentioned in table 4).
Round 2
Reviewer 2 Report
The authors answered all the questions and comments of the reviewer correctly and I think that this paper is now ready for publication.